# Sensors Based on Amino Group Surface-Modified CNTs

**Natalia Boroznina *** **, Irina Zaporotskova, Sergey Boroznin and Evgeniy Dryuchkov**

Institute of the Priority Technologies, Volgograd State University, 100 Universitetskii prospect, Volgograd 400062, Russia; irinazaporotskova@gmail.com (I.Z.); boroznin@volsu.ru (S.B.); dryuchkov.evgeniy@yandex.ru (E.D.)
* Correspondence: n.z.1103@mail.ru; Tel.: +7-909-391-02-44

**Abstract:** This article discusses the possibility of the fabrication of a highly sensitive sensor based on single-walled carbon nanotubes surface modified with functional amino groups ($-NH_2$). The sensor potential for detection of alkali (sodium, lithium, and potassium) metals was investigated. The results of computer simulation of the interaction process between the sensor and an arbitrary surface of the modified tube containing atoms of the studied metals are presented. The calculations were carried out within the framework of the density functional theory (DFT) method using the molecular cluster model. It has been proved that surface-modified ammonium carbon nanotubes show high sensitivity for the metal atoms under study.

**Keywords:** carbon nanotubes; sensor properties; functional group; amino group

---

## 1. Introduction

The last decades have witnessed dramatic advances in the fields of electronic engineering and nanoelectronics. This was mainly due to the use of carbon nanotubes, a unique material with new mechanical, electronic and magnetic properties (CNT) [1–4]. In addition to many distinguishing characteristics such as high sorption activity [5] and unique conductive properties [6], nanotubes acquire unique electronic properties when various atoms are adsorbed on their surface, which makes it possible to use them as chemical and biological sensors [7–13].

In order to increase the sorption activity of CNTs, the modification of the surface or boundaries of nanotubes by various functional groups is often used [14]. As a modifying group for a carbon nanotube, researchers often use a carboxyl functional group, which contributes to the creation of reactive sites at the boundaries or side walls of a nanotube. Thus, in Sun's article [15], it was found that carbon nanotubes modified with a carboxyl group shows sensitivity to CO with a detection limit of 0.00001, whereas unmodified (clean) nanotubes did not react to the presence of this gas [16,17]. Besides carboxyl groups, CNTs are often modified by amino groups. Tsai and co-authors investigated the sensitivity of the single-walled carbon nanotubes modified by amino functional group to carbon dioxide [18].

Amino-functionalized carbon nanotubes are potentially useful as sensors for other chemical species, including metals, as well as metal ions that are part of salts and alkalis. We have previously conducted research into carbon nanotubes that were boundary modified by carboxyl [17,18], amino and nitro groups [19,20]. It was found that boundary modification of CNTs by the above mentioned functional groups results in increased sensitivity which arises due to charge transfer [21]. However, the surface modification of CNTs is of no less interest. The paper discusses the possibility of CNTs modification by amino group with the view to study its properties as sensor for alkali metals (sodium, lithium, potassium). Earlier, research was carried out to explore the possibility of non-modified CNTs

for sensing alkali metals (lithium and potassium) [6]. It was defined that between the CNT surface and alkali atoms a rather strong bond is formed at an average distance of 1.6 Å. This means that when a CNT is used as an element of the sensor system for sensing alkali metals, it is necessary to provide conditions to break bonds form between alkali metal atoms and the CNT surface. Otherwise, the sensor system will stop functioning following the first contact with atoms it is fabricated to detect and extra time will be required to restore its functionality. It is suggested that this drawback should be overcome by using amino group surface modified nanotube.

The present paper discusses the possibility of a single-layer nanotube surface modification with an amino group with the view to use it for fabrication of a sensor that detects atoms of alkali metal (sodium, lithium, and potassium). Theoretical calculations were carried out by using the commonly applied Density Functional Theory (DFT) method described by Koch and Holthausen [22].

## 2. The Main Principles of the DFT Method

According to Density Functional Theory, the properties of a many-electron system including energy, can be defined by using an electron density functional. The system is described by electronic density as $\rho(\mathrm{r})$:

$$\rho(\mathrm{r}) = \int \ldots \int |\Phi_e|^2 \, d\sigma_1 d\sigma_2 \ldots d\sigma_N \tag{1}$$

where $\Phi_e$ is the many-electron wave function of the system, $\sigma_i$ is the set of spin and spatial coordinates of electrons, $N$ is the number of electrons. Thus, $\rho(\mathrm{r})$ is a function of only three spatial coordinates r of the point at which $\rho(\mathrm{r})$ gives the probability of detecting any of the electrons of the molecule [22].

If any property of the ground state of a molecule can be expressed in terms of $\rho$, then the electron energy in the DFT is:

$$E[\rho] = T[\rho] + V_{en}[\rho] + V_{ee}[\rho] \tag{2}$$

where $T[\rho]$ is the kinetic energy, $V_{en}[\rho]$ is the potential energy of electron-nuclear interactions, $V_{ee}[\rho]$ is the energy of electron-electron interactions, which can be written as:

$$V_{ee}[\rho] = V_{Coul}[\rho] + V_{xc}[\rho], \tag{3}$$

where $V_{Coul}[\rho]$ is the energy of the Coulomb interaction of electrons, and $V_{xc}[\rho]$ is the exchange-correlation energy.

The functionals $T[\rho]$, $V_{en}[\rho]$ and $V_{Coul}[\rho]$ can be found exactly [22]. For the exchange-correlation potential $V_{xc}[\rho]$, the exact representation is not known and there are a large number of models for its description. DFT is used with various functionals and one of the most popular is B3LYP, a hybrid functional that includes three components of the exchange functional (exact Hartree-Fock exchange operator, Becke functional and Slater functional, and the correlation part is a combination of the Lee-Yang-Parr functional (LYP) and Vosko-Vilka-Nusar (VWN). A feature of this approach is that the three exchange components are taken with weighting factors selected on the basis of comparison with experimental data. As a result, the approach takes on the characteristics of a semi-empirical method. It turns out that its accuracy in most cases is significantly higher than in the case of methodologically "pure" functionals. Apparently, this is a consequence of the fact that the exchange energy is nonlocal in nature and any attempts to reduce it to local functionals lead to errors. The Hartree-Fock exchange makes it possible to take this nonlocality into account. Therefore, in the presented theoretical study, the B3LYP functional was used within the framework of the density functional theory.

For the calculations, we used a Pople's valence double-zeta basis set 6-31G basis set (defined for the atoms from H to Zn) in which the valence atomic orbitals (AO) are composed of two parts—the inner, more compact, and the outer, more diffuse. Abbreviation 6-31G means that six primitive Gaussian functions are used to describe the core orbitals (non-valent electrons), and the valence s- and p-orbitals are divided into a compact part consisting of three Gaussian functions, and a diffuse part, which is represented by a single Gaussian function [23]. Calculations using such a basis set

reproduce, with sufficient accuracy, various energy parameters. To prove this, the tests were done using a large amount (299) of experimental energy values for molecules [23], such as 148 heats of formation, 85 ionization potentials, 58 electron affinity values and seven proton affinity values. It is believed that all these experimental results are known with an accuracy of 1 kcal/mol and higher. It was found that the calculations gave a standard deviation of 1.02 kcal/mol from these values. That is, the calculated accuracy turned out to be comparable with experimental accuracy [24–26]. Currently, many theoretical studies of carbon nanotubes using a B3LYP hybrid functional and a 6-31G base set have been performed [27–31]. Therefore, we can assume that our choice of the calculation method and the basis set is correct.

## 3. A Study of Carbon Nanotube Surface Modification by an NH$_2$ Amino Group

To study the process of a single-walled CNT surface modification by a NH$_2$ amino group, we considered the model of nanotube of type (6,0) molecular cluster that contained 96 carbon atoms. Loose bonds at the edge of the cluster were closed by hydrogen pseudo-atoms. The amino group was attached to a surface carbon atom located approximately in the middle of the cluster in order to exclude the influence of edge atoms. Figure 1 shows a model of the system. We modeled the mechanism of NH$_2$ group attachment to a selected atom on the nanotube surface with an increment of 0.1 Å along the perpendicular line drawn through the selected atom C to the nanotube axis. Then, we performed calculations by the DFT method and constructed an energy interaction curve (Figure 2). The analysis of the curve revealed that the value of interaction energy between a CNT and an amino group is 0.69 eV. This suggests bond formation between the nanotube and NH$_2$ amino group at a distance of 1.65 Å. Thus, the value of interaction energy indicates the possibility to fabricate a chemically active sensor based on amino group surface modified CNTs.

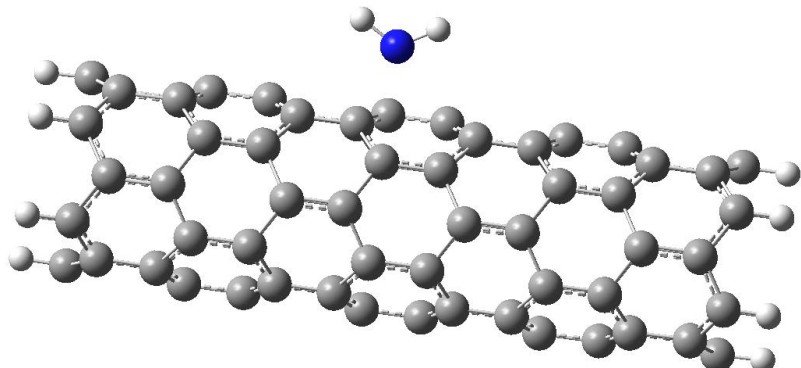

**Figure 1.** The CNT model with a surface functionalized by an amino group.

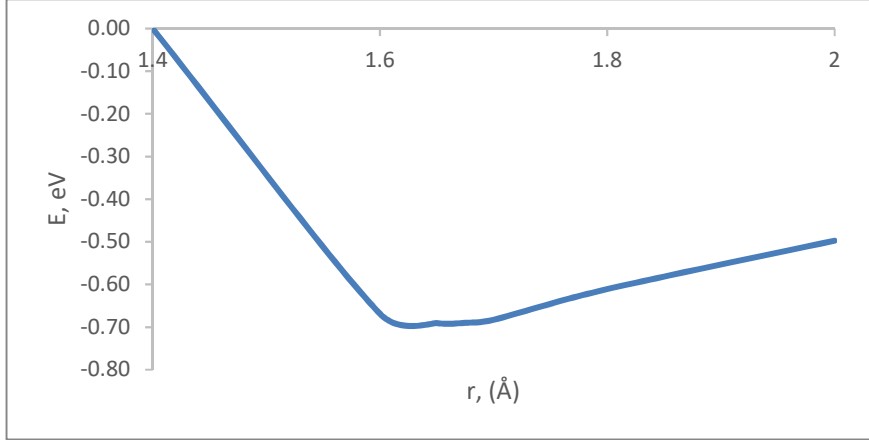

**Figure 2.** The CNT and an amino group interaction energy curve.

We analyzed the charge distribution (electron density distribution) in the system "CNT + $NH_2$" and found that the charges on the N atom of the functional group was +0.4; the charge on the hydrogen atom was −0.2. It should be noted that the charge calculation scheme proposed by R. Mulliken is used [32]. The Mulliken population analysis is based on the contributions of atomic orbitals to the molecular orbitals of the system; the population matrix is constructed by multiplying the elements of the density matrix and the overlap matrix. In this case, electronegativity is not taken into account. The electron density of an atom is determined by the sum of the squares of the decomposition coefficients of a molecular orbital in the atomic orbitals of a given atom plus half the overlap cloud of the orbitals of a given atom with a neighboring atom. The value of the charge on the N atom indicates that when the amino group is attached to the CNT surface electron transfer from N atom to the surface of CNT takes place. In other words, the sensor mechanism is implemented as a result of which additional charge carrier appears in the resulting nanosystem that ensures conductivity in the system.

Further, we investigated the band structure of the sensor "CNT + $NH_2$" Analysis of the band gap revealed that the system possesses semi-conducting properties ($\Delta E_g$ = 1.1 eV), and the electron of the amino-group serves as the charge carrier.

## 4. A Study of the Interaction Process between CNT–$NH_2$ System and Atoms and Ions of Alkali Metals

Further, we studied the interaction process between lithium, sodium, potassium atoms and edge hydrogen atoms of the amino group surface-modified carbon nanotube. The process was modeled incrementally by alkali metal (Na, K, Li) atoms (ions) to an H atom of the functional group. As a result of the calculations, surface profiles were constructed that show potential energy of the CNT–$NH_2$–Me system (Figure 3). Analysis of the results revealed that the binding process of selected metal atoms to the hydrogen atom of the group is barrier-free. In addition, due to a sufficiently large distance of interaction (of about 2.2 Å) between an amino group and alkali metals (it corresponds to the minimum on the energy curve), this interaction can be qualified as a weak van der Waals interaction, which enables multiple uses of the sensor as no chemical bonds are formed between the sensor and an alkali metal atom. Previous research considered conditions for desorption of atoms or ions to resume the sensor activity, among which ultraviolet radiation [33] or sensor heating [34] were proposed. A sensor modified by an amino group will be able to register the change in the value of the Schottky barrier between the electrodes of the sensor device and the CNT–$NH_2$ system. Table 1 shows the results of calculations for the interaction between selected atoms and a surface-modified CNT.

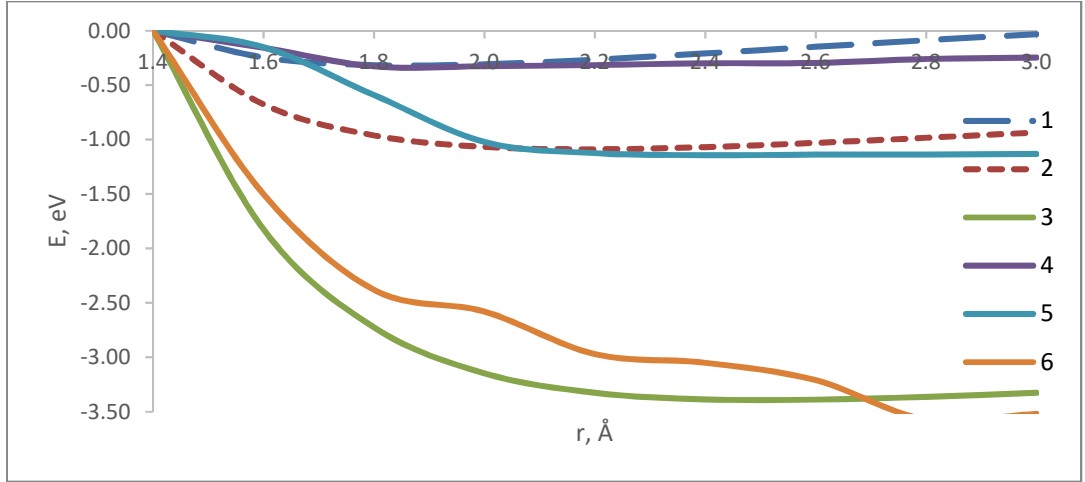

**Figure 3.** Energy curves of interaction between an amino-group modified nanotube with atoms and ions: 1 Li atom; 2 K atom; 3 Na atom; 4 $Li^+$ ion; 5 $Na^+$ ion; 6 $K^+$ ion, dependent on the distance between atoms/ions of metals and the hydrogen atom of the group.

**Table 1.** The main characteristics of Li, Na, K atom binding to an edge H atom of an amino group that modifies the nanotube surface: $r_{int}$—interaction distance between a metal atom and H atom of the functional group, $E_{int}$—a corresponding interaction energy; Q—charges on metal atoms.

| Atomic Bonds | $r_{int}$, Å | $E_{int}$, eV | Q |
|:---:|:---:|:---:|:---:|
| Li–H | 1.8 | −0.32 | +0.40 |
| Li$^+$–H | 1.8 | −0.33 | +0.66 |
| Na–H | 2.2 | −0.41 | +0.52 |
| Na$^+$–H | 2.2 | −1.12 | +0.79 |
| K–H | 2.6 | −0.66 | +0.73 |
| K$^+$–H | 2.8 | −3.57 | +0.91 |

Analysis of the charges on the atoms of the complex revealed that electron density is transferred from alkali atoms to an amino group surface-modified carbon nanotube. This transfer causes a change in the electrical properties of the complex as the number of charge carriers increases. The band gap in the system "CNT + NH$_2$" + atom/ion of metal decreases as compared to $\Delta E_g$ of the sensor system "CNT + NH$_2$" and becomes equal to 0.5 eV. This occurs due to the appearance of energy levels in the band gap, which is characteristic of the sensor nanosystem, to which the atomic potassium orbitals contribute.

## 5. Modeling the Scanning Process of the Nanotube Surface That Contains Atoms and Ions of Alkali Metals

A simulation of the scanning process of an arbitrary surface containing atoms and ions of lithium, sodium, or potassium was performed, and the activity of the surface-modified CNT with respect to selected metals was calculated. The process was modeled by an increment approach of a metal atom/ion to the functional group along a straight line drawn parallel to the surface of the nanotube (Figure 4). The line lay at a distance that was equal to the interaction distance for an atom/ion of a metal with the hydrogen atom of the amino-group modified nanotube (see Table 1). An analysis of the energy interaction curves constructed as a result of calculations (Figure 5) found that the modified nanotube is chemically sensitive for selected metals: the curves have a characteristic minimum that indicates the formation of stable interaction between the element and the CNT–NH$_2$ system. Table 2 shows the results of a calculation for sensory interaction between an amino group that modifies the nanotube surface and metal atoms. It was found that the energy minima were located in the middle between the hydrogen atoms of the group. This proves that the sensory sensitivity of the sensor based on surface-modified carbon nanotube is provided by the entire system with a functional amino group.

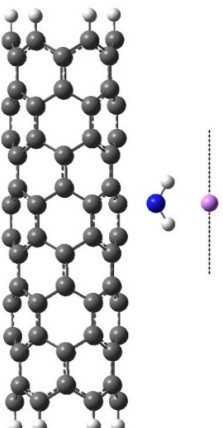

**Figure 4.** A scan of an arbitrary site on the surface modified carbon nanotube (6,0) that contain a Li atom (a pink color ball of a larger size); the dotted line indicates the route of Li atom movement in relation to an amino group functionally modified nanotube.

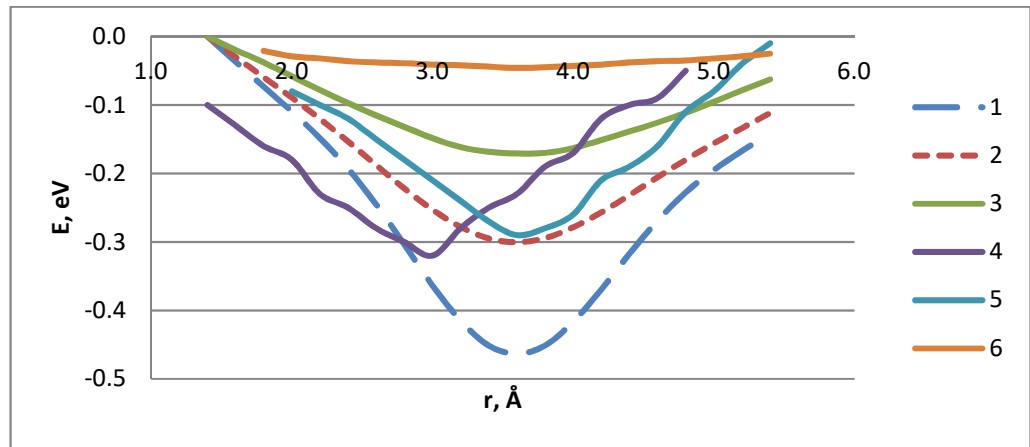

**Figure 5.** Energy curves of the scanning process of an arbitrary surface containing with atoms and ions: 1 Li atom; 2 K atom; 3 Na atom; 4 Li$^+$ ion; 5 Na$^+$ ion; 6 K$^+$ ion, surface-modified system "CNT + NH$_2$" The 1 Å point is located under the hydrogen atom of the amino group, to which the metal atom moves.

**Table 2.** Sensor interaction energy E$_{s\text{-int}}$ between amino group surface modified nanotube system and Li, Na, K atoms and ions.

| Atomic Bonds | E$_{s\text{-int}}$, eV |
|:---:|:---:|
| K–H | −0.46 |
| K$^+$–H | −0.32 |
| Na–H | −0.30 |
| Na$^+$–H | −0.29 |
| Li–H | −0.17 |
| Li$^+$–H | −0.05 |

## 6. Conclusions

The studies have shown that an amino group surface-modified carbon nanotubes are sensitive to alkali metal atoms and potentially, may be useful as sensors for alkali atom detection with acceptable sensitivity. The simulation results proved that the obtained "CNT + NH$_2$" system enables multiple uses as a probe for a specific set of elements. The interaction process is calculated by measuring the potential energy in a sensor system based on a nanotube modified by a functional group. The revealed change in potential energy is due to existence of additional charge carriers in the system that is the result of the system interaction with metal atoms. This interaction causes changes in its electronic properties. The sensor device obtained in this way will have specific selectivity determined by the energy of its interaction with various elements as the system responds in different ways to the presence of alkali metals. Surface-modified nanotubular systems can be used to create sensors in the form of active plates, the surface of which is coated with modified carbon nanotubes. In this case, the response of the system is provided by the total response of the entire surface to the presence of atoms or ions of alkali metals, which may be present in the form of solutions, salts and other alkali metal-containing compounds. Sensors fabricated by the proposed mechanism will be able to detect super small amounts of substances, up to the presence of atoms, which opens up a wide range of applications in biology, chemistry, medicine, etc.

**Author Contributions:** Data curation, E.D.; Formal analysis, S.B.; Investigation, N.B.; Project administration, I.Z.

**Acknowledgments:** The reported research was funded by Russian Foundation for Basic Research, grant No. 18-42-343009.

**Conflicts of Interest:** The authors declare no conflict of interest.

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
