# Peer review of "Sensors Based on Amino Group Surface-Modified CNTs"

_chemosensors, doi:10.3390/chemosensors7010011_

Reviewer 1 Report

Boroznina and her co-workers have performed model calculations for the interaction of an amino-group modified model carbon nanotube and alkali metals and alkali metal ions using DFT calculations. The manuscript is not well-written, and the content is beyond the scope of the journal of chemosensors. It would better fit to the scope of a quantum-chemistry or molecular modelling journal. These calculations do not have any close relevance to a real sensor. Unfortunately, even the quantum-chemistry part of the manuscript is badly written, not acceptable due to the lack of information usually expected in this field. Authors of this manuscript are seemingly not aware of limitations of the quantum-chemical method they use. A few other comments: details of calculations are not provided. Chapter 2 is trivial. More than this is expected from a university student; therefore, it cannot be part of a scientific publication. Basis set is not provided in the manuscript. It is not clear what kind of charges are presented in the manuscript. It is not clear how the band gap was calculated. 

Author Response

Answers:

1. " ..even the quantum-chemistry part of the manuscript is badly written, not acceptable due to the lack of information usually expected in this field"

Answer:  The reviewer does not write, what information did he expect to receive on the topic of the article. Quantum-chemical calculations allow us to obtain a variety of characteristics and parameters. Our article presents only the information needed to solve the problem stated in this paper.

2. " details of calculations are not provided"

Answer:  We would like to clarify what details of the calculations, according to the reviewer, it was necessary to submit? The paper describes the cluster model of the nanosystem, the calculation method, the potential. If the reviewer tells us what details he is interested in, we will be happy to answer his specific question.

3. “Chapter 2 is trivial. More than this is expected from a university student; therefore, it cannot be part of a scientific publication."

Answer: Chapter 2 was added in response to the remark of the previous reviewer, who asked to include the main provisions of the theory of DFT in the article.

4. "It is not clear what kind of charges are presented in the manuscript."

Answer: The charge describes the distribution of electron density in the system therefore it is given in units of electron charge.

5. “…It is not clear how the band gap was calculated.” 

Answer: It is found that the levels of the molecular orbitals are combined into zones. The width of the forbidden zone (so-called band gap ΔEg) is determined in our case as the difference between the energies of the high occupied molecular orbital and lower unoccupied molecular orbital.

Reviewer 2 Report

This is a good theoretical paper, but it could be improved by several changes

- Although the relative error of the experimental method was asserted to be lower than other methods, a more extensive discussion of error associated with the results would be useful.

-  The sections of the paper are incorrectly numbered (there are two section 3's leading to the other sections having incorrect numbering)

- Not sure why the paper has extensive areas highlighted in yellow

- When referring to references, the authors use the phrase "In [reference number]," in two places. It is customary to use the names of the authors or some identifier with the reference number, not just the number 

- The authors should be consistent in their use of decimal points...in most cases a decimal point (.) is used but in several cases a comma (,) is used (Tables 1&2)

- The Table 1 caption does not mention the column "Charge on metal atoms" nor is a unit given 

- English text needs to be edited. Suggested improvements are below:

Line 10: Insert the word "a" between the words "of" and "highly"

Line 11: Change the order of the words in the sentence to read ""single-walled carbon nanotubes surface modified with functional amino groups (-NH2)."

Line 16: Substitute the word "proven" for the word "proved"

Line 21: Begin the sentence with the word "The" so it reads "The last decades..." 

Line 22: Insert the word "a" after "nanotubes,"

Line 31: Change "nanotube" to "nanotubes"

Line 32: Delete the word "the" after the word "Besides"

Line 33: Change the word "group" to "groups" in both locations in the sentence

Lines 35 - 38: The phrasing of these sentences is somewhat awkward...a possible re-phrasing is..."Amino-functionalized carbon nanotubes are potentially useful as sensors for other chemical species including metals as well as metal ions." If re-phrased in this way, the sentence should be combined with the paragraph that currrently begins on line 39.

Line 39: Insert the words "that were" between the words "nanotubes" and "boundary"

Line 47: Insert the word "a" before "CNT"

Line 48: Change "bond formation" to "bonds formed"

Line 50: Change "extra time is required" to "extra time will be required"

Line 53: Insert "a" between the words "of" and "sensor"

Line 54: Insert the word "the" between "using" and "commonly"

Line 55: Lengthen the phrase to be "applied Density Functional Theory (DFT) method described by Koch and Holthausen [22]"

Line 58: Insert the word "an" between "using" and "electron"

Line 74: Eliminate the phrase "It is said that" and combine the sentences such as..."DFT is used with various functionals and one of the most popular is B3LYP,.."

Line 101: Insert the word "the" between "words," and "sensor"

Line 117: Change "metals" to "metal"

Line 124: Change "atoms" to "atom"

Line 126: Change the first part of the sentence to "A sensor modified by an amino group..."  

Line 132: Insert the word "the" between "as" and "number"

Line 158: Insert the word "a" between "of" and "calculation"

Line 180: The phrase after the word "and" is confusing. A possible re-phrasing is..."potentially, may be useful as sensors for alkali atom detection with acceptable sensitivity."

Line 191: The last word in the sentence needs clarification...a possible re-phrasing could be "in the form of solutions, salts and other alkali metal-containing compounds."

Author Response

Reviewer 2.

Answers:

1. “Although the relative error of the experimental method was asserted to be lower than other methods, a more extensive discussion of error associated with the results would be useful.”

Answer: Text added to the article (lines 85-88):

For the B3LYP hybrid functional, good convergence of experimental and theoretical results for carbon systems has been proved. The error of theoretical calculations for them is no more than 1% in terms of geometrical parameters (interatomic bonds and bond angles) and total energies.

2. “The sections of the paper are incorrectly numbered (there are two section 3's leading to the other sections having incorrect numbering)”

Answer: The corrections made.

3. “Not sure why the paper has extensive areas highlighted in yellow”

Answer: These are corrections made to the article on the recommendation of the reviewers.

4. “When referring to references, the authors use the phrase "In [reference number]," in two places. It is customary to use the names of the authors or some identifier with the reference number, not just the number”

Answer: Text are added:

Line 30: “In the Sun’s article [15], 

Lines 33-35 “Tsai and co-authors investigated the sensitivity of the single-walled carbon nanotubes modified by amino functional group to carbon dioxide [16]. 

5. “The authors should be consistent in their use of decimal points...in most cases a decimal point (.) is used but in several cases a comma (,) is used (Tables 1&2)”

Answer: The corrections made

6. “The Table 1 caption does not mention the column "Charge on metal atoms" nor is a unit given”

Answer: The charge describes the distribution of electron density in the system, therefore it is given in units of electron charge.

The corrections made, text are added (lines 151-152):  “Q - charges on metal atoms, units of electron charge”

7. “English text needs to be edited. Suggested improvements are below”

Answer: We are very grateful to the reviewer for help in correcting the text! The text was corrected.

8. Line 10: Insert the word "a" between the words "of" and "highly"

The corrections made, line10

9. “Line 11: Change the order of the words in the sentence to read ""single-walled carbon nanotubes surface modified with functional amino groups (-NH2)."

The corrections made

10. Line 16: Substitute the word "proven" for the word "proved"

The corrections made

11. Line 21: Begin the sentence with the word "The" so it reads "The last decades..." 

The corrections made

12. “Line 22: Insert the word "a" after "nanotubes,"  

The corrections made

13. Line 31: Change "nanotube" to "nanotubes" 

The corrections made

14. “Line 32: Delete the word "the" after the word "Besides" 

The corrections made, line 33 now

15. “Line 33: Change the word "group" to "groups" in both locations in the sentence”

The corrections made

16. “Lines 35 - 38: The phrasing of these sentences is somewhat awkward...a possible re-phrasing is..."Amino-functionalized carbon nanotubes are potentially useful as sensors for other chemical species including metals as well as metal ions." If re-phrased in this way, the sentence should be combined with the paragraph that currently begins on line 39”.  

The corrections made, lines 36-37

17. Line 39: Insert the words "that were" between the words "nanotubes" and "boundary"  

The corrections made, Now line 38

18. “Line 47: Insert the word "a" before "CNT"”

The corrections made, line 46

19. “Line 48: Change "bond formation" to "bonds formed"  

The corrections made, line 47

20. “Line 50: Change "extra time is required" to "extra time will be required"  

The corrections made, line 49

21. “Line 53: Insert "a" between the words "of" and "sensor"  

The corrections made

22. “Line 54: Insert the word "the" between "using" and "commonly" 

The corrections made.

23. “Line 55: Lengthen the phrase to be "applied Density Functional Theory (DFT) method described by Koch and Holthausen [22]"

The corrections made.

24. “Line 58: Insert the word "an" between "using" and "electron"  

The corrections made, line 59.

25. “Line 74: Eliminate the phrase "It is said that" and combine the sentences such as..."DFT is used with various functionals and one of the most popular is B3LYP,.." 

The corrections made, line 75

26. “Line 101: Insert the word "the" between "words," and "sensor" 

The corrections made, line 107 now

27. “Line 117: Change "metals" to "metal"  

The corrections made, line 122 now

28. Line 124: Change "atoms" to "atom" 

The corrections made, line 129

29.”Line 126: Change the first part of the sentence to "A sensor modified by an amino group..."   

The corrections made, line 131

30. “Line 132: Insert the word "the" between "as" and "number"  

The corrections made, line 137

31. “Line 158: Insert the word "a" between "of" and "calculation"  

The corrections made, line 164 now

32. “Line 180: The phrase after the word "and" is confusing. A possible re-phrasing is..."potentially, may be useful as sensors for alkali atom detection with acceptable sensitivity."

The corrections made, line 186-187

33. “Line 191: The last word in the sentence needs clarification...a possible re-phrasing could be "in the form of solutions, salts and other alkali metal-containing compounds."     

The corrections made, line 197-198

Reviewer 3 Report

The research article entitled “Sensors based on amino group surface modified CNTs” explored the possibility of a CNT – ammonia system for alkali metals detection. The conclusions in the article are backed up by the computational simulation with no experimental verification. The basis of the simulation, however, is controversial: the CNT-NH2 attachment mentioned in the article is not amino group functionalized SWCNTs, which is a critical scientific argument failure. The amino group functionalization on carbon dangling bonds from CNT edge plane has been extensively studied and well defined. Another argument failure in this article is the undefined matrix, whether the simulation is studied in a liquid phase or gaseous phase. Overall, quite lots of improvements and revision are required before to be considered for publication.

Author Response

Reviewer 3.

The research article entitled “Sensors based on amino group surface modified CNTs” explored the possibility of a CNT – ammonia system for alkali metals detection. The conclusions in the article are backed up by the computational simulation with no experimental verification. The basis of the simulation, however, is controversial: the CNT-NH2 attachment mentioned in the article is not amino group functionalized SWCNTs, which is a critical scientific argument failure. The amino group functionalization on carbon dangling bonds from CNT edge plane has been extensively studied and well defined. Another argument failure in this article is the undefined matrix, whether the simulation is studied in a liquid phase or gaseous phase. Overall, quite lots of improvements and revision are required before to be considered for publication.

Answers:

1. “The conclusions in the article are backed up by the computational simulation with no experimental verification.”

Answer:

Unfortunately, there are currently no experimental studies on sensory activity related to metals, their salts or alkalis. We hope that the theoretical studies performed will be useful for stimulating experimental work in this area.

2. Answer:

We disagree with the reviewer’s opinion that: “the CNT-NH2 attachment mentioned in the article is not amino group functionalized SWCNTs, which is a critical scientific argument failure". It is not clear what this opinion is based on? Unfortunately, the reviewer does not give any arguments as to why he believes that a nanotube with an amino group attached to the surface is not considered functionalized. This terminology is generally accepted.

3. Answer:

The reviewer notes the following: "The amino group functionalization on carbon dangling bonds from CNT edge plane has been extensively studied and well defined". We agree with this opinion. But in our article we are talking about the surface functionalization of the nanotube by an amino group, when the group joins the carbon atom of the surface, and not the dangling carbon bonds on the edge of the CNT plane.

4. ”Another argument failure in this article is the undefined matrix, whether the simulation is studied in a liquid phase or gaseous phase”.

Answer:

The simulation does not take into account the presence of the medium. An ideal case of the presence of only a modified nanotube (CNT + amino group) and an atom or metal ion is considered. This is a general principle of theoretical modeling.

Reviewer 4 Report

The paper deals with the possibility to fabricate highly sensitive sensors based on surface-modified single-walled carbon nanotubes for the detection of alkali atom and ions. The approach is only based on computational analysis and there is not any validation of the results. The topic is interesting, but the authors should try to make any efforts to validate their computational analysis by supporting it with some kind of measurements. 

Author Response

Reviewer 4.

The paper deals with the possibility to fabricate highly sensitive sensors based on surface-modified single-walled carbon nanotubes for the detection of alkali atom and ions. The approach is only based on computational analysis and there is not any validation of the results. The topic is interesting, but the authors should try to make any efforts to validate their computational analysis by supporting it with some kind of measurements. 

Answer:

Unfortunately, there are currently no experimental studies on sensory activity related to metals, their salts or alkalis. We hope that the theoretical studies performed will be useful for stimulating experimental work in this area.

Round  2

Reviewer 1 Report

Unfortunately, the manuscript has not improved at all. It is just the opposite due to wrong new statement (see below). My personal impression after reading authors` comments is that they blindly used a quantum-chemistry software without knowing the theoretical background and the applicability and limits of quantum-chemical methods to solve chemical problems. The content of the manuscript is not closely relevant to the scope of Chemosensors. The manuscript is badly written and information are useless without specifying exactly how they are obtained.  

--- Not only the quantum-chemical method, but the basis set used for calculation must be specified. Total energy, bond lengths, charge, and many other parameters depend on basis set, not only on the method applied. In addition, proper basis set must be selected for solving a chemical problem.  

--- Using a quantum-chemistry software there is always a default built in for running the program, but it is often not the method of choice. Considering charges, the default is the worst, its application must be avoided for many systems. Authors do not understand reviewer`s question, what indicates that they are not experts in this field, but it is clear for anyone who is experienced: Have you calculated Mullican charges, Hirschfeld charges, charges from natural orbital analysis, or anything else? Stating that charges were calculated does not mean anything.

--- The new sentence in the manuscript, viz. “For the B3LYP hybrid functional, good convergence of experimental and theoretical results for carbon systems has been proved. The error of theoretical calculations for them is no more than 1% in terms of geometrical parameters (interatomic bonds and bond angles) and total energies”, is clearly wrong. It also lacks the literature reference. Not surprisingly, because it is not true. How the authors of this manuscript want to gain experimental total energies??????????????????? If the theoretical and experimental geometrical parameters (bond lengths and angles) agree, there is clearly something wrong. Quantum-chemical methods provide equilibrium structures (viz. structure at the potential energy minimum) for a rigid isolated system. Experimentally derived structures are the result of averaging molecular motions. If it is derived by crystallography, it reflects also solid phase effects. Authors should know what they compare, and how experimental structures were obtained. The physical meaning of quantum-chemically derived structure and any experimental structure is different.   

Author Response

Please find response in the attachment.

Reviewer 3 Report

Accept

Author Response

Thank you.

This manuscript is a resubmission of an earlier submission. The following is a list of the peer review reports and author responses from that submission.

Round  1

Reviewer 1 Report

This manuscript showcases the potential of primary amine-modified single carbon nanotube (SWNT)  for sensing alkali metal atom and ions. Unfortunately, most of the details of calculation lacks out even though this manuscript focuses on the DFT simulation of amine-modified SWNT and alkali metal atoms. I cannot approve the manuscript in this form.

Comment 1: The potential change between unmodified SWNT and alkali ion needs to be calculated or properly cited at least to show the necessity of SWNT-NH2 system rather than SWNT by itself.

Comment 2: In section 4, the vertical distance from amine to the moving line of alkali metal is missing. It needs to be clarified.

Comment 3: In section 3, potassium presented the highest interaction energy through vertical trajectory, but in section 4, lithium showed the highest interaction through parallel trajectory. The inverted order of alkali metal atoms for interaction energy needs to be addressed.

Comment 4: The more details regarding DFT calculation will be helpful for readers.

Comment 5: The interaction energy curve for alkali metal ions needs to be added in the manuscript or supporting information.

Comment 6: In line 68-70, I cannot understand what grounds conclude into the sentence “the sensor will be destroyed”. It needs more detailed explanation.

Comment 7: There are lots of typos and incorrect phrases. It needs to be revised for publication.

Reviewer 2 Report

In the manuscript, the authors investigate, from a theoretical point of view, the possibility to use carbon nanotubes functionalized by amino group (NH2) to detect alkali metals (sodium, lithium and potassium).

They perform the calculation by using DFT method and show the energy interaction curves for each case.

In principle, the matter of the manuscript is interesting, but the results here shown are poor and of little interest for the scientific community focused on the study of carbon-based sensor.

Why do the authors show only the energy interaction curve?

The DOS and the electronic structure at the interface between CNT-NH2 and CNT-NH2-metal are lacking. Why? All DFT results should be presented and the discussion should be deepened and enlarged.

In my opinion, the manuscript must be rejected.